# Systemic, Mucosal Immune Activation and Psycho-Sexual Health in ART-Suppressed Women Living with HIV: Evaluating Biomarkers and Environmental Stimuli

**DOI:** 10.3390/v15040960

**Published:** 2023-04-13

**Authors:** Parni Nijhawan, Anna Carraro, Serena Vita, Cosmo Del Borgo, Eeva Tortellini, Mariasilvia Guardiani, Maria Antonella Zingaropoli, Fabio Mengoni, Vincenzo Petrozza, Luciana Di Troia, Immacolata Marcucci, Blerta Kertusha, Maria Cristina Scerpa, Ombretta Turriziani, Vincenzo Vullo, Maria Rosa Ciardi, Claudio Maria Mastroianni, Raffaella Marocco, Miriam Lichtner

**Affiliations:** 1Department of Public Health and Infectious Diseases, Sapienza University of Rome, 00185 Rome, Italy; 2Infectious Diseases Unit, Santa Maria (SM) Goretti Hospital, Sapienza University of Rome, 04100 Latina, Italy; 3INMI IRCCS Lazzaro Spallanzani, via Portuense, 00149 Rome, Italy; 4UOC of Pathology, Department of Medical-Surgical Sciences and Bio-Technologies, Sapienza University of Rome Polo Pontino, I.C.O.T, 04100 Latina, Italy; 5Department of Andrology and Pathophysiology of Reproduction, Santa Maria (SM) Goretti Hospital, 04100 Latina, Italy; 6Hematology Unit, Santa Maria (SM) Goretti Hospital, AUSL Latina, 04100 Latina, Italy; 7Department of Molecular Medicine, Sapienza University of Rome, 00161 Rome, Italy; 8Department of Neurosciences, Mental Health, and Sense Organs, NESMOS, Sapienza University of Rome, 00189 Rome, Italy

**Keywords:** WLWH, HW, ART, genital tract, mucosal immune response, cytokines, FSFI, GAD test

## Abstract

Gender medicine is now an approach that can no longer be neglected and must be considered in scientific research. We investigated the systemic and mucosal immune response in a population of women living with HIV (WLWH) who were receiving successful ART and the sexual and psychological repercussions of HIV infection on the women’s health. As control group, healthy women (HW) matched for age and sex distribution, without any therapy, were included. In summary, our study highlighted the persistence of immune-inflammatory activation in our population, despite virological suppression and a normal CD4 cell count. We found a hyperactivation of the systemic monocyte and an increase in inflammatory cytokine concentrations at the systemic level. The analysis carried out showed a significantly higher risk of HPV coinfection in WLWH compared to HW. Furthermore, our data revealed that WLWH have a profile compatible with sexual dysfunction and generalized anxiety disorders. Our study underlines that patients living with HIV should be evaluated by multidisciplinary teams. These findings also support the idea that more and different immunological markers, in addition to those already used in clinical practice, are needed. Further studies should be carried out to clarify which of these could represent future therapy targets.

## 1. Introduction

HIV is a global epidemic, causing infections in both men and women disproportionately [1,2]. However, in the past few years, several studies have been carried out that highlight an increase in the number of women living with HIV (WLWH) globally, evidencing that adolescent girls and middle-aged women represent a majority in the high-risk category for contracting HIV and other related co-infections [3,4,5,6,7].

It is not well-known whether the genital tract is a site of chronic inflammation in WLWH as it has been described for the gut; however, it certainly represents a reservoir for HIV from the blood to the local viral replication site [8,9].

A complex network of immune protection resides throughout the female reproductive tract, and this network constitutes what is known as mucosal immunity, which comprises both innate and adaptive components. These two systems work in tandem and through integrated interactions to provide protection at both mucosal and systemic levels [10,11].

Cytokines and chemokines also play a key role in eliciting an innate immune response. Anti-inflammatory cytokines such as IL-10 and pro-inflammatory cytokines such as IL-6, IL-8, IL-12, TGF-beta, and IL-1 beta drive the development and functioning of the local immune cells and increase in the female genital tract in the case of an acute HIV infection [12,13,14]. Prior studies suggested that increased levels of inflammatory cytokines in the female genital tract could promote local viral replication and/or recruit immune target cells in the site, leading to increased HIV shedding [12,14].

With the advent of antiretroviral therapy (ART), WLWH are now living longer and have an improved quality of life; however, diminished sexual function among these women has been recently associated with more severe HIV-related symptoms and a decrease in adherence to ART [15,16]. Although some studies have failed to detect associations between sexual function and biomarkers of HIV-associated disease, a few have reported that WLWH experience lower levels of sexual interest, pleasure, and satisfaction when compared with non-infected women [17]. Many factors affect the female sexual function, including biological, psychological, physiological, socio-cultural, and couple relationship factors [15,16]. Defined as “persistent, recurrent problems with sexual response, desire, orgasm or pain that causes distress and strains relationships,” female sexual dysfunction (FSD) affects the lives of many women worldwide [15,16].

The aim of the study was to investigate the systemic and mucosal immune response in a population of WLWH receiving successful ART in accordance with environmental stimuli.

The study was divided as follows:The evaluation of clinical parameters and immune responses, considering the CD4, CD8, and NK cell count and cytokines (sCD163, sCD14, IL-1 Beta, IL-6, CxCl10, and IL-8) in the plasma and vaginal lavage (VL) of both WLWH on ART and healthy women (HW).Screening for viruses, such as Cytomegalovirus (CMV) and Human Papillomavirus (HPV), and bacteria causing genital tract infection.The evaluation of female sexual dysfunction, generalized anxiety disorder, and female hormones to describe the sexual health status of WLWH on ART.

## 2. Materials and Methods

### 2.1. Ethics Statement

This study was approved by the Ethics Committee of Sapienza University (Protocol number 129.17) once it was established that the research would be conducted in compliance with the indications of ethics and health protection, as established by the Ministry of Health of the Italian Government.

Each subject gave written informed consent for participation in the study.

### 2.2. Study Cohort and Sampling

WLWH receiving effective ART and undergoing routine follow-ups at the Infectious Diseases Unit of the S.M Goretti Hospital of Latina, Italy, were enrolled. As a control group, HW matched for age and sex distribution and without any therapy were included.

Blood samples were collected using ethylenediaminetetraacetic acid (EDTA) tubes and spray-coated silica and polymer-gel-containing vacutainers (BD Biosciences, Franklin Lakes, NJ, USA). Peripheral whole blood samples collected in EDTA tubes were centrifuged for 10 min at 1800 rpm to separate the cellular fraction and plasma. The plasma was then removed from the cell pellet and stored at −20 °C.

Serum samples were obtained by centrifuging peripheral whole blood samples collected in spray-coated silica and polymer-gel-containing vacutainers at 3000 rpm for 10 min and stored at −20 °C.

VL samples were collected in sterile test tubes by dilating the vaginal canal with the help of specula and washing with 10 mL of sterile physiological solution. The VL samples were stored at −80 °C prior to being analyzed. Finally, women were subjected to a cervical swab for HPV-DNA. 

### 2.3. Estimation of Lymphocytes and Cytokines

Blood samples were acquired using the MACSQuant Flow Cytometer (Miltenyi Biotec, Bergisch Gladbach, Germany) for the characterization of lymphocyte subpopulations (absolute and percentage CD3+, absolute and percentage CD4+, absolute and percentage CD8+, CD4/CD8 ratio, and absolute and percentage NK cells and B cells).

A flow cytometric estimation of lymphocytes was performed using antibodies from Milteny Biotec and BD Biosciences. Specifically, lymphocyte subpopulations were assessed using BD MultitestTM 6-color TBNK (FITC-labeled CD3 clone SK7; PE-labeled CD16, clone B73.1, and CD56, clone NCAM 16.2; PerCP-CyTM5.5-labeled CD45, clone 2D1; PE-CyTM7-labeled CD4, clone SK3; APC-labeled CD19, clone SJ25C1; APC-Cy7-labeled CD8, clone SK1) as previously described in these articles [18,19].

Proinflammatory cytokines and soluble biomarkers of immune activation were determined by using Enzyme Linked Immunosorbent Assay (ELISA) kits.

A Quantikine^®^ R & D System ELISA kit (RayBiotech Life, Peachtree Corners, GA, USA) was used for the determination of IL-1β, sCD163, and sCD14 (with sensitivity <1 pg/mL, 0.058–0.613 ng/ml and <125 pg/mL respectively).

A Diaclone^®^ ELISA kit (Diaclone SAS Besancon Cedex, France) was used for the determination of IL-6 and IL-8 (with sensitivity 0.81 pg/mL and 29 pg/mL respectively).

### 2.4. Estimation of Sexual Health and Psicological Well-Being

To track sexual well-being, the Female Sexual Function Index questionnaire (FSFI), which covers a total of 19 questions, was administered at the time of examination [15,20]. A score of 26.55 or lower is an index of FSD [15,20]. 

To measure the psychological state, the Generalized Anxiety Disorder (GAD) test was used [15]. A score above 50 points refers to a possible initial to moderate form of depression [15].

### 2.5. Coinfections Screening

An enzyme immunoassay with microparticles in chemoluminescence (CMIA) from Abbott Architect i1000SR^®^ (Abbott, Abbott Park, IL, USA) was used for the determination of Hepatitis B Virus (HBV) markers (HBsAg, HBcAb, and HBsAb) specific to anti-Hepatitis A Virus (HAV) and anti-Hepatiti C Virus (HCV) immunoglobulins.

A CMV ELITe MGB^®^ real-time PCR kit (Thermo Fisher Scientific 168 Third Avenue) Waltham, MA, USA) was utilized for the determination of CMV-DNA in serum and vaginal lavage.

The investigation of pathogens responsible for the sexually transmitted diseases (STDs) (Ureaplasma urealiticum, Ureaplasma parvum, Mycoplasma genitalium, Mycoplasma hominis, Neisseria gonorrhoeae, Chlamydia trachomonis, and Trachomonis vaginalis) and HPV-DNA on a cervical swab was carried out by multiplex real-time PCR Allplex TM STI Essential Assay and Anyplex TM II HPV28 Detection, Seegene^®^ (Thermo Fisher Scientific 168 Third Avenue Waltham, Waltham, MA, USA), respectively.

### 2.6. Quantitative Determination of Sexual Hormones

The quantitative determination of testosterone, progesterone, and estradiol was carried out by means of an immune-enzymatic assay with microparticles in chemiluminescence (CMIA) from Abbott Architect^®^ (Abbott, Abbott Park, IL, USA).

### 2.7. Statistical Analysis

Statistical analyses were performed using GraphPad Prism, v.9, for macOS. A two-tailed *p* ≤ 0.05 was considered statistically significant. Data are represented as median with an interquartile range (IQR).

The nonparametric comparative Mann–Whitney test and the nonparametric Kruskal–Wallis test with Dunn’s post-test were used for comparing the medians between groups. Baseline clinical and demographic characteristics were compared by a chi-square test. Correlations between quantitative data were assessed using the non-parametric Spearman test. Linear correlation was evaluated using the regression test.

## 3. Results

### 3.1. Participant Characteristics

Overall, a total of 58 subjects were enrolled in the study after having signed the informed consent.

Of these participants, 24 were WLWH receiving effective ART and 34 were HW. Among the HW, 3 were receiving EP therapy and 31 were HW without any therapy. 

The clinical parameters considered during the study and the type of ART therapy are summarized in Table 1.

### 3.2. Characterization of Immune Cells and Subpopulation of Lymphocytes

Table 2 summarizes the lymphocyte subpopulation in the two groups.

The WLWH demonstrated a good immunological response to therapy; indeed, no significant differences in the absolute count and percentage of CD4 T cells in WLWH and HW were observed. However, CD8 T lymphocytes were significantly higher in the WLWH compared to the HW in terms of absolute count and percentage (*p* = 0.002 and *p* < 0.0001, respectively) (Table 2).

Conversely, NK cells were significantly lower in the WLWH compared to the HW in terms of absolute count and percentage, (*p* = 0.007 and *p* = 0.03, respectively); the same result was observed for the CD4/CD8 ratio (*p* < 0.0001) (Table 2).

No significant differences were found for the B cell population (Table 2).

### 3.3. Estimation of Cytokines and Soluble Biomarkers of Blood in the Plasma and Vaginal Lavage

The WLWH showed statistically significant higher sCD163 plasma levels (*p* = 0.008), as well as significantly higher IL-6, IL-1β, and IL-8 plasma levels compared to the HW (*p* < 0.0001, *p* = 0.02 and *p* = 0.03, respectively) (Figure 1A). No significant differences were found in the levels of sCD14 and CxCL10 (Figure 1A).

Considering vaginal lavage, IL-6 levels were significantly lower in the WLWH compared to the HW (*p* = 0.008) (Figure 1B).

### 3.4. CMV Co-Infection

CMV-DNA was measured in all participants through serum and VL samples. In the WLWH, 4.3% (1/24) had positive results, while in the HW, CMV-DNA was undetectable in all cases (100%). The VL samples were negative for CMV-DNA in all participants.

Of the WLWH, 95.6% were positive for serum anti-CMV-IgG versus 79% of the HW. Moreover, the levels of anti-CMV IgG were significantly higher in the WLWH compared to the HW (160 [5.0–1030.0] IU/mL and 80.30 [0.0–137] IU/mL, respectively, *p* < 0.0001) (Figure 2).

### 3.5. Human Papilloma Virus (HPV)

Overall, a significantly higher prevalence of HPV infection was found in the WLWH compared to the HW (*p* = 0.0476) (Figure 3A).

In the WLWH, 62.5% (15/24) were positive for HPV-DNA, with a total of 16 different HPV genotypes (Figure 3A,B). In WLWH, high-risk HPV genotypes were present in 25% (6/24) of cases and an association of “high-risk” with “low-risk” was found in 21% (5/24) of cases. “Low risk” genotypes were present in 17% (4/24). Therefore, 46% (11/24) of the women had an HPV infection related to a high risk of developing precancerous and cancerous lesions.

In the HW, HPV-DNA was detected in 29% (10/34) of cases, with a total of 12 different genotypes (Figure 3A,B).

“High-risk” genotypes were exclusively found in 15% (5/34) of the women, while 3% (1/34) were positive for the “low-risk” genotype. A combination of "high-risk" and "low-risk" genotypes was found in 12% (4/34) of the HW. Therefore, the "high-risk" genotypes were detected in 27% (9/34) of them. 

### 3.6. Sexually Transmitted Microorganisms and Commensal Microorganisms

Of the WLWH, 62.5% (15/24) tested positive for an STM. *Ureoplasma parvum* was the most common organism found (46%), followed by *Mycoplasma hominis* (17%) and *Ureoplasma urealiticum* together with *Tricomonas vaginalis* (4%) (Table 3). A triple infection was found, with *U. urealiticum* and *T. vaginalis* specifically isolated and associated with *U. parvum* infection in the same subject. This group tested negative for other causative agents of STD (Table 3). 

In the HW, the most common STM was *U. parvum* (21%) (Table 3).

Regarding commensal microorganisms, 17% of the WLWH had positive results. Specifically, the most common were *Escherichia coli* and *Streptococcus agalactiae* (8%) (Table 3).

In the HW, *E. coli* and *S. agalactiae* were the most common (8%), followed by *Proteus mirabilis* (6%) (Table 3).

### 3.7. Estimation of Sexual Hormones in Different Phases of Menstrual Cycle

In the WLWH, significantly lower levels of testosterone (0.185 (0.1–1.15) nmol/L vs. 0.24 (0.12–0.4) nmol/L, respectively, *p* = 0.04) and progesterone (0.17 (0.1–17) ng/mL vs. 2.7 (0.1–15) ng/mL, respectively, *p* = 0.03) were observed compared to the HW, while no significant differences were found between them in levels of estradiol (Figure 4). Overall, 10/24 WLWH and 18/34 HW were in the follicular phase, 8/24 and 10/34 were in the luteal phase, and 6/24 and 6/34 were in menopause.

### 3.8. Evaluation of Sexual Health and Psychological Well-Being

The FSFI includes different domains such as desire, excitation, lubrification, orgasm, satisfaction, and pain. Overall, the desire and lubrification domains were significantly lower in the WLWH compared to the HW (3.0 (0.0–6.0) and 4.2 (1.8–6.0), *p* = 0.004; 3.3 (0.0–6.0) and 5.1 (0.0–6.0) *p* = 0.01, respectively).

The GAD test score revealed that 20% (5/24) of WLWH were suffering from an initial to moderate form of depression versus 15% (5/34) of the HW.

Generalized anxiety seemed to persist equally in the two groups (Figure 5B).

### 3.9. Representation of Inflammatory, Immunological, and Sexual Health Markers in WLWH and HW

Trends of biomarkers in 24 WLWH and 34 HW were represented (immune cells and subpopulation of lymphocytes, cytokines and soluble biomarkers of blood in the plasma and vaginal lavage, CMV IgG seropositivity, sexual hormones, and health) by a radar chart in Figure 6. 

As previously reported, the WLWH showed statistically significant higher sCD163 IL-6, IL-1β, and IL-8 plasmatic levels (*p* = 0.008, *p* < 0.0001, *p* = 0.02 and *p* = 0.03, respectively) with respect to the HW, as well as a higher absolute CD8 cell count (*p* = 0.002) and anti-CMV IgG levels (*p* < 0.0001). On the contrary, the NK absolute cell count (*p* = 0.007), CD4/CD8 ratio (*p* < 0.0001), vaginal IL-6 levels (*p* = 0.008), testosterone levels (*p* = 0.04), progesterone levels (*p* = 0.03), as well as FSFI, were lower in the WLWH compared to the HW.

### 3.10. Correlations between Measured Parameters

In the WLWH, a positive correlation between sexual hormones and plasmatic pro-inflammatory biomarkers, specifically between plasmatic CxCl10 and estradiol levels (r = 0.5, *p* = 0.01), was observed (Figure 7A). Correlations between sexual health in terms of FSFI and sexual hormones were also found. In detail, the WLWH displayed a positive correlation between FSFI and testosterone levels (r = 0.56, *p* = 0.02) and FSFI and estradiol levels (r = 0.51, *p* = 0.02) (Figure 7B,C). Correlations between sexual health, psychological wellness, and immuno-virologic factors were also considered. Negative correlations were found between the status of sexual health (FSFI) and anxiety (Z-index) (r = −0.49, *p* = 0.03), as well as between years of HIV diagnosis and FSFI (r = −0.43, *p* = 0.05) (Figure 7D,E).

In the HW, plasmatic sCD163 (r = 0.3, *p* = 0.05) revealed a positive correlation with estradiol levels (Figure 7F). The age of the individuals (r = −0.5, *p* = 0.01) was in a negative correlation with testosterone levels (Figure 7G).

## 4. Discussion

HIV infection continues to be a major global health issue, with an estimated 36.7 million people living with HIV (PLWH) worldwide.

High-risk, vulnerable populations include female sex workers, transgender individuals, individuals belonging to ethnic minorities, and individuals with disabilities [21]. Heterosexual transmission is the primary mode of HIV infection for the vast majority of women around the world, and critical gaps remain in the understanding of immuno-biological mechanisms of HIV infection [21]. 

Mucosal surfaces account for most of the transmission of HIV and play a crucial role in HIV acquisition; the spreading of the virus depends on its ability to cross a mucosal barrier [22,23,24]. 

Although the introduction of antiretroviral therapy (ART) has led to significant reduction in morbidity and mortality in PLWH, viral replication persists and accelerated aging and some other comorbidities may accompany this therapy [25,26]. 

It is not well-known whether the genital tract is a site of chronic inflammation in WLWH, and the impact of ART on immune activation and inflammation, memory marker distribution, and T cell exhaustion in the genital tract are not well-studied. Therefore, we investigated the systemic and mucosal immune response in a population of WLWH receiving successful ART, in accordance with environmental stimuli, taking into account that gender medicine is now an approach that can no longer be neglected and must be considered in scientific research. We then highlighted the sexual and psychological repercussions that HIV infection has on women’s health.

In our study, we found no significant differences in the CD4 T cell count when comparing WLWH to HW, indicating the effectiveness of ART and an apparent successful immune recovery. Moreover WLWH generally have a higher a CD4 T cell count than men living with HIV (MLWH) [27]. We observed a higher CD8 T cell count and a lower CD4/CD8 ratio compared to the HW, although higher than 1. Furthermore, we found a lower NK cell count, in accordance with the fact that chronic HIV infection maintains an altered population distribution and functional capacity of NK cells, despite the complete recovery of the T-helper population [28].

Concerning cytokine production, we found that the WLWH showed statistically significantly higher plasma levels of sCD163 compared to the HW, as well as higher IL-1β, IL-6, and IL-8 plasma levels. These results are in line with previous studies that showed a residual activation of the monocyte/macrophage line despite successful ART, in particular of sCD163 [29]. Higher levels of IL-6, IL-8, and IL-1 β have been observed in WLWH, and IL-1 β tends to normalize in patients receiving ART; however, the values are often not comparable to those of the healthy population [30]. 

While the systemic level of proinflammatory monocytes/macrophages was higher in the WLWH, the IL-6 and IL-8 levels in VL were significantly lower in this population compared to the HW, indicating a possible defective production at this level.

Finally, no significant differences were found in the levels of sCD14 and CxCL10 in either sample type. sCD14 has found to be increased in PLWH without good viro-immunological control and is correlated with higher bacterial translocation and mortality [31]. Like sCD14, CXCL10 levels were comparable to those of the HW, confirming what is already known from the literature: the existing data have shown that once ART is initiated, CXCL10 values return to normal values [32]. 

Our data, in line with other studies, support the idea that it is necessary to evaluate multiple immunological markers that have not yet been validated in clinical practice to obtain a more complete and detailed profile of a WLWH.

In a similar work, Cheret et al. evaluated the inflammatory status of man living with HIV (MLWH) by measuring proinflammatory cytokines such as IL-6 and IP-10 and markers of monocyte activation (sCD163 and CD14) in blood. They found that the monocyte activation markers correlated negatively with the semen HIV-RNA load in recently infected men, in accordance with the fact that early innate immunity contributes to the control of HIV replication in semen [33].

Since herpes virus coinfections, especially CMV coinfection, have been proposed as other key cofactors in sustaining immune activation in PLWH, we performed a parallel screening for CMV shedding and HPV-DNA [34]. In the general population, the worldwide seroprevalence for CMV is estimated to be 40 to 90%, with higher prevalence in PLWH [35]. In line with the literature, even in our WLWH, the prevalence of IgG for CMV was higher compared the HW (95.6% vs. 79%), with anti-CMV IgG levels significantly higher in in this group compared to the HW.

Only one sample from the WLWH group was found positive for CMV-DNA with an extremely low dosage, confirming that viral shedding in ART-treated women is not so common.

In our study, using cervico-vaginal swabs, we also performed a search for HPV as a known causative agent of cervical cancer, which is an AIDS-defining condition.

We observed a higher percentage of WLWH who were positive for HPV compared to the HW. This finding is in line with the idea that HPV persistence is dependent on CD4 but especially on NK cells and that high-risk genotypes maintain a state of immune activation with hyper-expression of the surface markers of T lymphocytes [36,37].

Similar to WLWH, MLWH have a higher incidence of HPV infection and other related coinfections, abnormal pap smears, and persistent HPV infection, leading to a higher risk of HPV-related cancers than in the general population [38].

We then highlighted the sexual and psychological repercussions that HIV infection has on women’s health. 

Among the WLWH, a higher proportion fell into the category of sexual dysfunction with respect to the HW. This result also reflects the data for the median number of days since the last instance of sexual intercourse.

On the contrary, an equal persistence of anxiety disorder was observed in the two groups.

It is interesting to underline that the values obtained from the FSFI questionnaire are inversely proportional to the outcome obtained with the GAD, suggesting a direct relationship between sexual well-being and psychological well-being [15,39].

There was also a clear correlation between the years that had passed since the diagnosis of HIV infection and the state of sexual dysfunction and generalized anxiety disorder. This data may therefore suggest that receiving a diagnosis of HIV infection implies a repercussion on psychological and sexual well-being and that patients must therefore be protected and accompanied by psychological support [15,39]. In summary, our study highlighted the persistence of immune-inflammatory activation in WLWH on ART, despite virological suppression and a normal CD4 cell count. In particular, we found a hyperactivation of the systemic monocytes and an increase in inflammatory cytokine concentrations at the systemic level, which also has repercussions at a vaginal level and could represent a risk factor for HPV-related diseases and could determine a higher susceptibility to other coinfections.

The analysis carried out showed a greater predisposition of PLWH to a co-infection with HPV, confirming the need to evaluate the patients belonging to HIV centers more carefully. However, the number of instances of sexual intercourse is lower in WLWH; therefore, this fact cannot be explained by greater promiscuity.

Many immunological and clinical parameters can help us in profiling a WLWH and supporting her throughout her life. Further studies are needed to investigate which parameters could be useful for consideration as immunological markers in addition to those already used in clinical practice and which of these could represent future therapy targets.

Finally, WLWH have pictures compatible with sexual dysfunction and generalized anxiety disorders, underlining the need for PLWH to be evaluated by multidisciplinary teams.

## Figures and Tables

**Figure 1 viruses-15-00960-f001:**
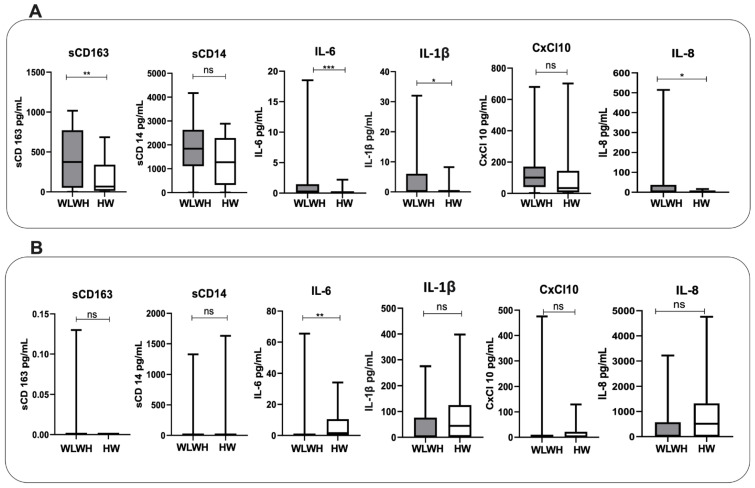
Concentration of proinflammatory biomarkers in WLWH and HW in plasma (**A**) and vaginal lavage (**B**). The differences between the two groups were evaluated using the nonparametric Mann–Whitney test. Data are shown as median (boxplots). Statistical significance (*p*) is reported in the graphics. *** *p* < 0.001; ** *p* < 0.01; * *p* < 0.05.

**Figure 2 viruses-15-00960-f002:**
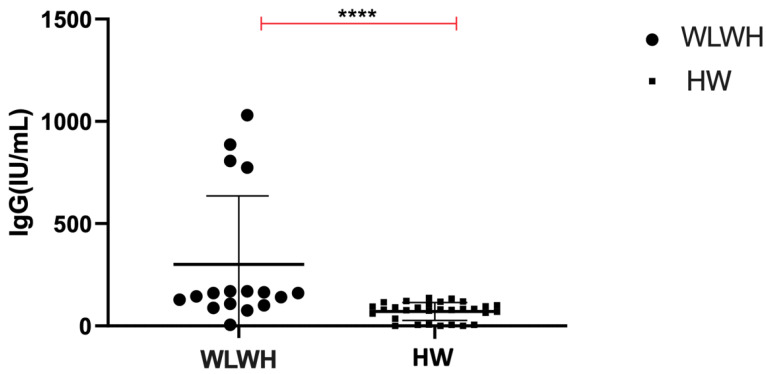
Quantitative dosage of anti CMV-IgG levels in WLWH and HW. The differences were evaluated using the nonparametric Mann–Whitney test. Data are shown as median (lines). Statistical significance (*p*) is reported in the graphics. **** *p* < 0.0001.

**Figure 3 viruses-15-00960-f003:**
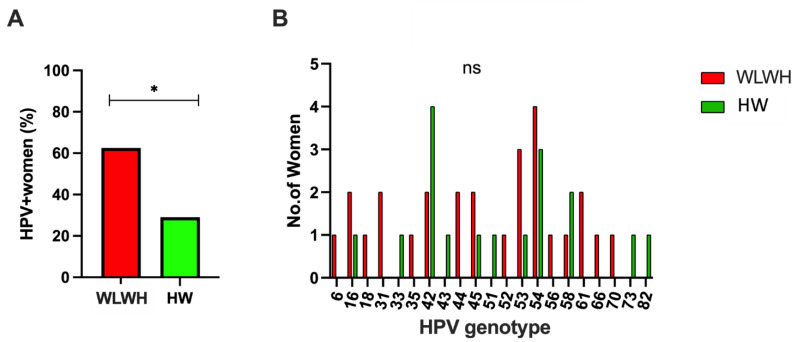
Screening for HPV in WLWH and HW. (**A**) The differences between the two groups were evaluated using the nonparametric Mann–Whitney test. Statistical significance (*p*) is reported in the graphics. * *p* < 0.05. (**B**) Different genotypes of HPV found in the study population (WLWH and HW).

**Figure 4 viruses-15-00960-f004:**
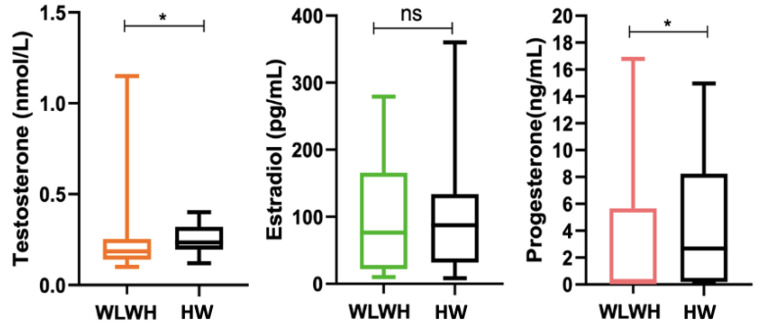
Estimation of sexual hormones in WLWH and HW. The differences between the two groups were evaluated using the nonparametric Mann–Whitney test. Data are shown as median (box plots). Statistical significance (*p*) is reported in the graphics. * *p* < 0.05.

**Figure 5 viruses-15-00960-f005:**
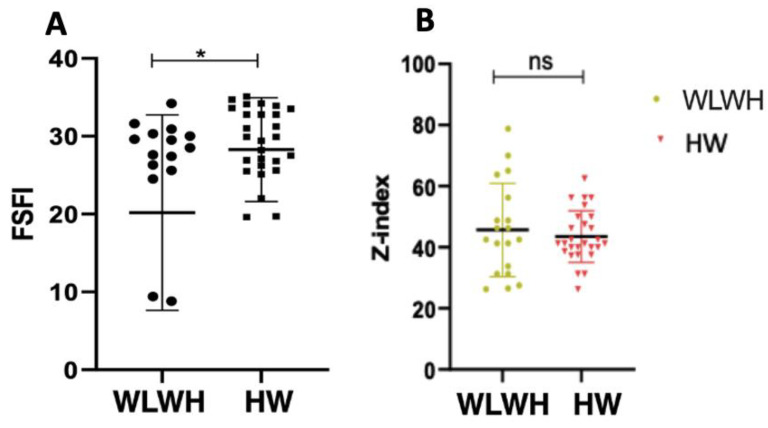
Sexual health and psychological well-being of the study population (**A**) Differences in overall FSFI scores in WLWH and HW. (**B**) Generalized anxiety disorder in WLWH and HW. Statistical significance (*p*) is reported in the graphics. * *p* < 0.05.

**Figure 6 viruses-15-00960-f006:**
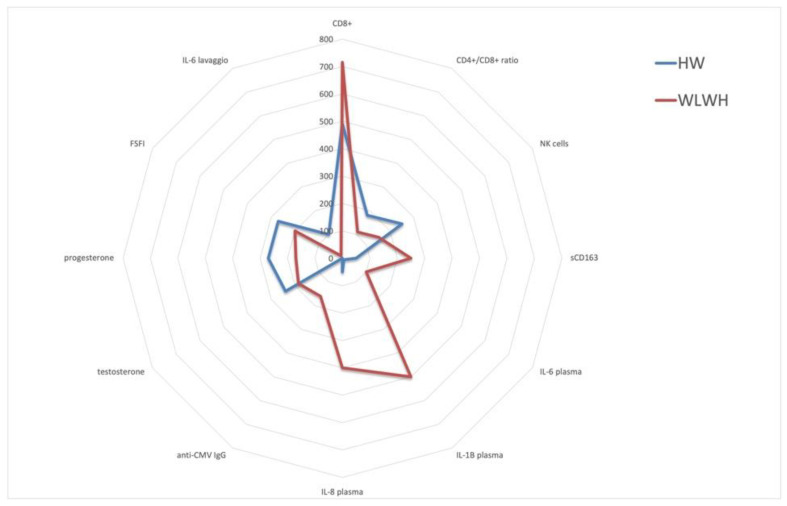
Representation of inflammatory, immunological, and sexual health markers: (CD4/CD8 ratio = cells/mm3; CD4+ = cells/mm3; NK cells = cells/mm3; sCD163 = soluble CD163; ng/mL; IL-6 = interleukin-6; pg/mL; IL-1beta = interleukin-1 beta; pg/mL IL-8 = interleukin-8 pg/mL; anti-CMV igG = IU/mL; testosterone; progesterone; FSFI) in plasma and vaginal lavage are represented by radar chart. Results are reported as estimated average variations. Radar plots were constructed scaling all variables in a range from 0% to 100% and plotting averages in WLWH and HW.

**Figure 7 viruses-15-00960-f007:**
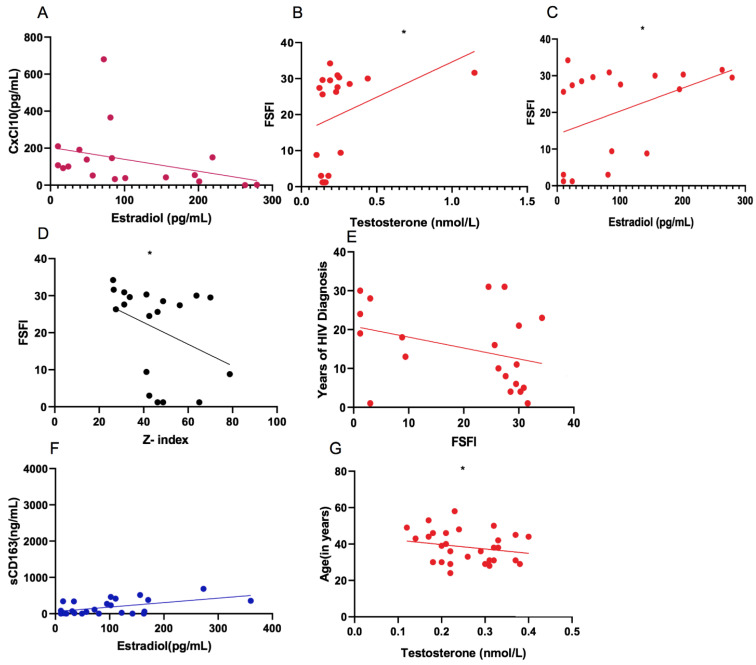
Correlations. (**A**) Positive correlation between CxCl10 and estradiol levels (r = 0.5, *p* = 0.01) in WLWH. (**B**) Positive correlation between FSFI and testosterone levels (r = 0.56, *p* = 0.02) in WLWH. (**C**) Positive correlation between FSFI and estradiol levels (r = 0.51, *p* = 0.02) in WLWH. (**D**) Negative correlation between FSFI and anxiety (Z-index) (r = −0.49, *p* = 0.03) in WLWH. (**E**) Negative correlation between years of HIV diagnosis and FSFI in WLWH (r = −0.43, *p* = 0.05). (**F**) Positive correlation between plasmatic sCD163 and estradiol levels (r = 0.3, *p* = 0.05) in HW. (**G**) Negative correlation between the age of the individuals and testosterone levels (r = −0.5, *p* = 0.01) in HW. All correlations were performed using the Spearman test. * *p* < 0.05.

**Table 1 viruses-15-00960-t001:** Baseline clinical and demographic characteristics of participants.

Parameter	WLWH (n = 24)	HW (n = 34)	*p*-Value
Age, median (IQR) years	41 (29–65)	39 (24–62)	ns
MAOIs n (%)	6 (25)	0 (0)	0.0033
EP therapy n (%)	5 (21)	3 (9)	ns
Nationality n (%)			
Italian n (%)	15 (63)	34 (100)	0.0001
Ukrainian n (%)	2 (8)	0 (0)	ns
African n (%)	7 (29)	0 (0)	0.0008
Education n (%)			
Elementary n (%)	2 (8)	2 (6)	ns
Lower Average n (%)	12 (50)	2 (6)	0.0001
Upper Average n (%)	10 (42)	6 (18)	0.04
Graduation n (%)	0 (0)	22 (64)	<0.0001
Diploma n (%)	0 (0)	2 (6)	ns
Partner n (%)	Yes: 15 (63)	Yes: 34 (100)	0.0001
Partner HIV + n (%)	Yes: 4 (17)	Yes: 0 (0)	0.01
Years of HIV Diagnosis	12 (10–31)	NA	-
HIV Zenith	54980 (1423–5,407,656)	NA	-
Occupation n (%)	11 (46)	30 (88)	0.0005
CDC Stages			
A1	3 (13)	NA	-
A2	7 (28)	NA	-
A3	5 (20)	NA	-
B1	0 (0)	NA	-
B2	3 (13)	NA	-
B3	3 (13)	NA	-
C3	3 (13)	NA	-
Smoking n (%)	11 (46)	16 (47)	ns
Alteration in PAP Test n (%)	10 (42)	4 (12)	0.09
CD4 Nadir Med (Min–Max)	266.5 (8–615)	NA	-
	ART Combination		
ABC + 3TC + DTG	1 (4)	NA	-
DRV/r + ETR	2 (8)	NA	-
ETR + ABC + 3TC	1 (4)	NA	-
LPV/r + 3TC	1 (4)	NA	-
TAF + FTC + BIC+	1 (4)	NA	-
TAF + FTC + EVG + COBI + DRV	1 (4)	NA	-
TAF + FTC + RPV	6 (26)	NA	-
TDF + FTC/DTG	1 (4)	NA	-
TDF + FTC + DTG+	2 (8)	NA	-
TDF + FTC + ELV/C	1 (4)	NA	-
TDF + FTC + RAL	2 (8)	NA	-
TDF + FTC + RPV	5 (22)	NA	-

WLWH: women living with HIV; HW: healthy women; PAP: Papanikolaou test; EP: estroprogestins; MAOIs: monoamine oxidase inhibitors; IQR: interquartile range; n: number; ART: antiretroviral therapy. Data are reported as median (interquartile range). Baseline characteristics were compared by chi square.

**Table 2 viruses-15-00960-t002:** Immune cells in WLWH and HW.

All Lymphocytes	WLWH: Median (Min–Max)	HW: Median (Min–Max)	*p*-Value
CD3+ mm^3^	1541 (831–3177)	1527 (909–2401)	0.4
CD3+ %	75.9 (61.1–90.4)	73 (7.7–85.9)	0.27
CD4+ mm^3^	767 (196–1665)	961.5 (482–1732)	0.15
CD4+ %	39.6 (15.1–48)	45.5 (33.9–61.9)	0.14
CD8+ mm^3^	715 (357–1521)	491 (303–853)	0.002
CD8+ %	36.1 (21.1–56)	24.4 (13.2–36.7)	<0.0001
CD4+/CD8+ ratio	1.11 (0.28–1.88)	1.81 (1.0–3.75)	<0.0001
NK cells mm^3^	153 (42–426)	251 (53–684)	0.007
NK cells %	8.6 (1.9–25.2)	12.9 (4.2–24.1)	0.03
B Lymphocytes mm^3^	204 (74–530)	223 (108–576)	0.5
B Lymphocytes %	12.3 (5.4–19.5)	10.25 (5.9–18.8)	0.55

WLWH: women living with HIV; HW: healthy women; NK: natural killer. The nonparametric comparative Mann–Whitney test was used to compare medians between the WLWH and HW.

**Table 3 viruses-15-00960-t003:** Sexually transmitted and commensal microorganisms in the study population.

Microorganism	WLWH (n = 24) n (%)	HW (n = 34) n (%)	*p*-Value
**Sexually transmitted microorganisms**			
*Ureoplasma parvum*	11 (46)	7 (21)	*p* = 0.0407
*Mycoplasma hominis*	4 (17)	1 (3)	ns
*Ureoplasma urealiticum*	1 (4)	1 (3)	ns
*Tricomonas vaginalis*	1 (4)	0 (0.0)	ns
*Nisseria gonnorhea*	0 (0.0)	0 (0.0)	ns
**Commensal microorganisms**			
*Escherichia coli*	2 (8)	2 (6)	ns
*Enterococcus faecalis*	0 (0.0)	1 (3)	ns
*Streptococcus agalactiae*	2 (8)	2 (6)	ns
*Streptococcus pyrogens*	0 (0.0)	1 (3)	ns
*Streptococcus hemolyticus*	1 (4)	0 (0.0)	ns
*Proteus mirabilis*	0 (0.0)	2 (6)	ns

Chi-square test was used to assess differences between the groups.

## Data Availability

The raw data supporting the conclusions of this article will be made available by the authors without undue reservation.

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
