# Peer review of "Systemic, Mucosal Immune Activation and Psycho-Sexual Health in ART-Suppressed Women Living with HIV: Evaluating Biomarkers and Environmental Stimuli"

_viruses, 2023, doi:10.3390/v15040960_

Round 1
Reviewer 1 Report
In the present paper, Nijhawan et al aim to assess systemic and mucosal immune activation and psycho-sexual health cART suppressed Women living with HIV (WLWH).
The authors performed a cross-sectional analysis on 24 WLLWH and 34 uninfected controls and analyze multiple markers (cytokines in plasma/vaginal fluid), STI (HPV, Ureaplasma urealiticum, Ureaplasma 121 parvum, Mycoplasma genitalium, Mycoplasma hominis, Neisseria gonorrhoeae, Chlamydia tra- 122 chomonis, Trachomonis vaginalis) as wella sexual and psychological parameters.
The paper is of potential interest, yet not acceptable in the present form for the follwing reasons:
i) a clear hypothesis is lacking; the introduction is way too long and includes too much information which distracts the reader from the aim of the study ii) results are presented as independent studies; no attempt is made to present the findings in an organic manner; iii) discussion is vague, long and does not convey a message.
Overall the paper is very difficult to follow. It is unclear what the real findings of the study are. In my opinion, the paper has to be re-written with a clear framing of the problem and hypothesis; results should be presented in a more organic fashion, following the authors' working hypothesis; discussion and conclusions should refer to the findings presented and highlight the significance of their findings.
Further, multiple errors are present in the text (e.g. ELISA described in two paragraphs; Figure 4: HD instead of HW) and English needs to be revised. I also thing tables need to be improved (too many).
Author Response
Referee 1
In the present paper, Nijhawan et al aim to assess systemic and mucosal immune activation and psycho-sexual health cART suppressed Women living with HIV (WLWH).
The authors performed a cross-sectional analysis on 24 WLLWH and 34 uninfected controls and analyze multiple markers (cytokines in plasma/vaginal fluid), STI (HPV, Ureaplasma urealiticum, Ureaplasma 121 parvum, Mycoplasma genitalium, Mycoplasma hominis, Neisseria gonorrhoeae, Chlamydia tra- 122 chomonis, Trachomonis vaginalis) as wella sexual and psychological parameters.
The paper is of potential interest, yet not acceptable in the present form for the following reasons:
- a clear hypothesis is lacking; the introduction is way too long and includes too much information which distracts the reader from the aim of the study.
- results are presented as independent studies; no attempt is made to present the findings in an organic manner.
- discussion is vague, long and does not convey a message.
Overall, the paper is very difficult to follow. It is unclear what the real findings of the study are. In my opinion, the paper has to be re-written with a clear framing of the problem and hypothesis; results should be presented in a more organic fashion, following the authors' working hypothesis; discussion and conclusions should refer to the findings presented and highlight the significance of their findings.
Further, multiple errors are present in the text (e.g. ELISA described in two paragraphs; Figure 4: HD instead of HW) and English needs to be revised. I also thing tables need to be improved (too many).
Our hypothesis is that, in WLWH receiving successful ART, despite a good immuno-virological status, the persistence of immune-inflammatory activation has repercussions also at psychological and sexual levels. Therefore, we investigated the systemic and mucosal immune response in a population of WLWH on ART and in parallel evaluated their sexual health, reinforcing the idea that WLWH should be evaluated by multidisciplinary teams and supported throughout their lives.
As suggested by the Referee, we shorten the introduction excluding the information that are not necessary to introduce the aim of the study. Furthermore, we reduced the number of tables present in the Result section (we removed Table 2 and table 4 Fig 4B) and tried to reformulate the discussion in a more organic manner, in order to better underline the significance of our findings.
We corrected the Materials and Methods section describing ELISA only in one paragraph (2.3 section Materials and Methods lines 114-120)
We corrected the labels in the Figure 4 with HW instead of HD.

Reviewer 2 Report
The MS by Nijhawan et al investigates some blood parameters together with some psychological aspects in a group of women living with HIV. The question of gender is interesting, rarely investigated in this field that often concentrates on MSM and needs to be addressed.
However, the data presented are very descriptive, the biological quantitative data not correlate to any specific biological function. The correlation with psychological parameters remains correlation and no causative relationship is established and no strategy to establish some is proposed.
Furthermore, detailed comparison with published data on men living with HIV could improve the MS.
Furthermore, although hormonal cycle time at sampling has been recorded for all patients, this parameter is not taken into account in the peripheral and local cytokine analyses, which might bias the data presented.
Main issues:
1- Methods should be detailed with ab concentration, etc… to be able to reproduce the experiments.
2- Where protease inhibitors included in the VL? Otherwise it might have affected the data.
3- The collected VL volume is certainly different between individual even if the same 10ml volume is injected. Therefore, cytokines quantification in VL must be normalized by total antibody or albumin or a constant factor. And correlation must be recalculated.
4- Statistics between WLWH and WH should be calculated in table 1, 4, fig 4B
5- Individuals should be split according to hormonal cycle when analyzing quantification of VL components, both cytokines and sexual hormones.
6- A multiparametric analysis between all cytokines in both compartment should be performed.
7- To get a better global understanding of the parameters analyzed, a multinomial logistic regression between measured parameters both biological and psychological should be performed and results discussed.
8- A comparison with literature on Men LWH should be made and discussed.
9- The consequence of the change in the cytokine measured on specific cells, immunity or other factors should be precisely discussed.
Additional points
1- Add in the abstract the type of individuals to which WLWH is compared
2- rephrase line 62 64: . there are three ideas in this sentence that do not define mucosal immunity. Which are not correctly linked. Innate and adaptive immunity exist also in systemic immunity. And interaction between mucosal and systemic immunity are both direction. Please provide a better definition of mucosal immunity, its compartimentalized nature.
3- L: 64-65Same with the following sentence. Better define innate immunity and specify it at both mucosal and systemic levels.
4- L 69: genital: male or female or both?
5- L74: are these diminished sexual functions associated to hormonal cycle?
6- Acronym need to be detailed at the first occurrence: HW, FSI, GAD …..
Author Response
The MS by Nijhawan et al investigates some blood parameters together with some psychological aspects in a group of women living with HIV. The question of gender is interesting, rarely investigated in this field that often concentrates on MSM and needs to be addressed.
However, the data presented are very descriptive, the biological quantitative data do not correlate to any specific biological function. The correlation with psychological parameters remains correlation and no causative relationship is established and no strategy to establish some is proposed.
Furthermore, detailed comparison with published data on men living with HIV could improve the MS.
Furthermore, although hormonal cycle time at sampling has been recorded for all patients, this parameter is not taken into account in the peripheral and local cytokine analyses, which might bias the data presented.
Main issues:
1- Methods should be detailed with ab concentration, etc… to be able to reproduce the experiments.
As suggested by the Referee we implemented the Methods in section 2.3, lines 108-120.
As previously described by Iannetta et al. in 2 articles (Iannetta, M.; et al, doi:10.3389/fimmu.2021.796482. and Iannetta, M et al doi:10.3389/fimmu.2021.796482.) we detailed the procedure for T-, B-, NK- Lymphocyte Assessment in Peripheral Blood. Specifically, lymphocyte subpopulations were assessed using BD MultitestTM 6-color TBNK (FITC- labeled CD3 clone SK7; PE-labeled CD16, clone B73.1, and CD56, clone NCAM 16.2; PerCP-CyTM5.5-labeled CD45, clone 2D1; PE-CyTM7-labeled CD4, clone SK3; APC-labeled CD19, clone SJ25C1; APC-Cy7-labeled CD8, clone SK1) and BD TrucountTM tubes were used for absolute count with a lyse-no-wash procedure (BD FACSTMLysing Solution). A stabilized blood sample (BDTM Multi-Check Control) was run for each working session.
Flow Cytometric estimation of lymphocytes was performed using antibodies from Milteny Biotec and BD Biosciences as described in Material and Methods section, paragraph 2.3, line 108-113. The ab concentration is calculated according to the datasheet. In particular we performed 50 tests with BD antibodies and 100 tests with Milteny antibodies
2- Where protease inhibitors included in the VL? Otherwise, it might have affected the data.
Protease inhibitors were not included in VL because vaginal fluid samples were collected and immediately stored at –80 C prior to analyze, as reported in other studies (Jaumdally et al., 2018. DOI:10.1038/s41598-018-30663-8)
3- The collected VL volume is certainly different between individual even if the same 10ml volume is injected. Therefore, cytokines quantification in VL must be normalized by total antibody or albumin or a constant factor. And correlation must be recalculated.
One of the most common approach to collect secretion samples from the female genital tract is the VL and this method is widely used to understand local immune processes that play a crucial role in many infections, diseases and reproductive processes (Jaumdally et al., 2018. DOI:10.1038/s41598-018-30663-8 and Zara et al., 2003, doi: 10.1136/sti.2003.005157).
In our opinion, a normalization of the cytokines measured was not necessary since during the procedure, performed by experienced gynecologists, particularly attention was given to the correct positioning of the speculum and to the initial volume of the lavage injected into the vagina (10 mL of sterile physiological solution) as well as to the suspension aspirated after the lavage (10 mL) and since the quantification was given in terms of relative concentration of each cytokine.
4- Statistics between WLWH and WH should be calculated in table 1, 4, fig 4B
As suggested by the Referee we calculated the statistics in table 1 line 161 and table 3 line 238 (Due to the elimination of table 3, table 4 has become table 3).
5- Individuals should be split according to hormonal cycle when analyzing quantification of VL components, both cytokines and sexual hormones.
As suggested by the Referee, we performed the analysis of VL components stratifying WLWH and HW according to hormonal cycle, and no statistically significant differences were found between the stratified subgroups.
6- A multiparametric analysis between all cytokines in both compartments should be performed.
7- To get a better global understanding of the parameters analyzed, a multinomial logistic regression between measured parameters both biological and psychological should be performed and results discussed.
In order to get a better global understanding of the parameters analyzed, as suggested by the Referee in the 6 and 7 points; we decided to represent trends of biological psychological parameters in WLWH and HW by radar plots (paragraph 3.9, figure 6). The plots were constructed scaling all variables (in a range from 0% to 100%) and plotting averages in WLWH and HW.
8- A comparison with literature on Men LWH should be made and discussed.
As suggested by the Referee, we made a comparison with literature on MLWH and discussed it in the Discussion section (lines 333-334; lines 358-363 and lines 380-382). Specifically, we added three articles in the bibliography (Chéret et al., 2017, doi: 10.1371/journal.pone.0180191; Ruel et al., 2011, doi:10.1093/cid/cir484; Rungmaitreee et al., 2022, DOI: 10.3390/vaccines10010118).
9- The consequence of the change in the cytokine measured on specific cells, immunity or other factors should be precisely discussed.
As suggested by the Referee we implemented the discussion section about the consequence of the change in the cytokine measured on immunity (lines 401-403).
Additional points
- Add in the abstract the type of individuals to which WLWH is compared
As suggested by the Referee, in lines 25-26, we corrected the abstract section and added type of individuals, specifically Healthy Women (HW) to which WLWH is compared.
- Rephrase line 62 64: there are three ideas in this sentence that do not define mucosal immunity. Which are not correctly linked. Innate and adaptive immunity exist also in systemic immunity. And interaction between mucosal and systemic immunity are both directions. Please provide a better definition of mucosal immunity, its compartmentalized nature.
- L: 64-65Same with the following sentence. Better define innate immunity and specify it at both mucosal and systemic levels.
As suggested by the Referee, we implemented the introduction section and better clarified the description of the mucosal immunity adding two articles and better define innate immunity (lines 47-51) (Caputo, V et al., doi:10.3389/fimmu.2023.1104423; Hickey, D.K et al., doi: 10.1016/j.jri.2011.01.005) .
L 69: genital: male or female or both?
As suggested by the Referee, we corrected the line 55 specifying “female genital tract”. Due to a shortening of the introduction section the line 69 has become line 55.
- L74: are these diminished sexual functions associated to hormonal cycle?
As previously described in introduction section, line 55-58 and line 67-69 many factors affect female sexual function, including biological, psychological, physio-logical, socio-cultural, and couple relationship factors. The FSFI test refers to the past 4 weeks and it is an assessment of diminished sexual function over time (FSFI test is illustrated in this link https://www.nva.org/wp-content/uploads/2015/01/FSFI-questionnaire2000.pdf ). However, in some women the test was repeated regularly in the clinic, and it was seen that it did not change much over time.
- Acronym need to be detailed at the first occurrence: HW, FSI, GAD.
As suggested by the Referee, we checked all the acronym in the text to be sure that they were detailed at the first occurrence.

Reviewer 3 Report
Yes we can proceed with the publication of the manuscript.
Author Response
Referee 3
Yes, we can proceed with the publication of the manuscript.
Thank you

Round 2
Reviewer 1 Report
The authors have extensively edited their manuscript.
However the introduction and discussion require further editing.
While I understand that CMV/HPV and other STI may be drivers of inflammation, the link between these factors and sexual dysfunction/anxiety needs some framing. The discussion is still too long: the authors tend to repeat the results section and fail to interpret their findings. They discuss in the following order: inflammtion, co-infections (HPV), sexual dysfunction and HPV again.